# A Metal Ion and Thermal-Responsive Bilayer Hydrogel Actuator Achieved by the Asymmetric Osmotic Flow of Water between Two Layers under Stimuli

**DOI:** 10.3390/polym14194019

**Published:** 2022-09-26

**Authors:** Wanting Dai, Xiaoyan Zhou, Huilong Guo

**Affiliations:** 1National Engineering Research Center for Healthcare Devices, Guangdong Key Laboratory of Medical Electronic Instruments and Polymer Material Products, Institute of Biological and Medical Engineering, Guangdong Academy of Sciences, Guangzhou 510316, China; 2Key Laboratory for Polymeric Composite and Functional Materials of Ministry of Education, GD HPPC Laboratory, School of Chemistry, Sun Yat-Sen University, Guangzhou 510275, China; 3Key Laboratory of Biomaterials of Guangdong Higher Education Institutes, Department of Biomedical Engineering, Jinan University, Guangzhou 510632, China

**Keywords:** hydrogel actuator, metal ion responsive, thermal responsive, self-deformation

## Abstract

Shape-morphing hydrogels have drawn great attention due to their wide applications as soft actuators, while asymmetric responsive shape-morphing behavior upon encountering external stimuli is fundamental for the development of hydrogel actuators. Therefore, in this work, bilayer hydrogels were prepared and the shrinkage ratios (L_A_/L_N_) of the AAm/AAc layer to the NIPAM layer immersed in different metal ion solutions, leading to bending in different directions, were investigated. The difference in the shrinkage ratio was attributed to the synergistic effect of the osmolarity difference between the inside and outside of the hydrogels and the interaction difference between the ion and hydrogel polymer chains. Additionally, under thermal stimuli, the hydrogel actuator would bend toward the NIPAM layer due to the shrinkage of the hydrogel networks caused by the hydrophilic–hydrophobic phase transition of NIPAM blocks above the LCST. This indicates that metal ion and thermal-responsive shape-morphing hydrogel actuators with good mechanical properties could be used as metal ion or temperature-controllable switches or other smart devices.

## 1. Introduction

Stimuli-responsive shape-morphing hydrogels are attracting increasing attention due to their shape-morphing responsiveness under external stimuli, such as heat, pH, light, force, ions, solvents, and electric or magnetic fields, making them suitable for a wide range of applications, including actuators, biomimetic actuators, artificial muscles, and soft robotics [1,2,3,4,5,6,7,8,9,10,11]. Significant work has been undertaken to develop varieties of stimuli-responsive hydrogels, such as thermo-responsive poly(N-isopropylacrylamide) (polyNIPAM) [12], salt-responsive polyVBIPS [13], and light-responsive graphene oxide (GO)-incorporated hydrogels [14,15]. However, hydrogels usually undergo isotropic volumetric expansion and contraction under stimuli, so the design and fabrication of anisotropic structures are fundamental for the development of hydrogel actuators [16].

To address this issue, various anisotropic structures, such as oriented structures [17], gradient structures [18], patterned structures [19,20], and bilayer structures, [12,14,16,21,22,23,24,25], have been investigated. Among them, bilayer structures are increasingly being favored by researchers for their ease of preparation and design diversity [26]. Bilayer hydrogels will undergo asymmetric responsive shape-morphing behavior upon encountering external stimuli, thus leading to macroscopic bending, stretching, and twisting. The asymmetric osmotic flow of water and interfacial adhesion between the two layers is crucial for hydrogel actuators [12,22]. In our previous work [27], a “C”- or “S”-shaped actuator was prepared by applying a thin layer of hydrophobic adhesive film to the hydrophilic hydrogel film according to specific shape designs. The actuator could bend toward the hydrophobic film due to the asymmetric osmotic flow of water, which was caused by differences in the hydrophilicity–hydrophobicity of the two layers. However, interfacial adhesion was formed by physical adhesion, which was not stable enough to undergo a long-term or repeated shape-morphing process.

It has been reported that a strong bilayer interface can be achieved through the formation of a semi-interpenetrating network structure at the interface of two hydrogel layers [13]. A semi-interpenetrating network can be formed by the in situ polymerization of a second hydrogel layer on the hydrogel layer prepared first. During the polymerization of the second layer, some active reactants will penetrate the first hydrogel network and then form a semi-interpenetrating network interface. NIPAM blocks may undergo phase transition and cause the shrinkage of the hydrogel networks at a temperature above the lower critical phase-transition temperature (LCST). Thus, during heating, the bilayer hydrogel will bend toward the NIPAM layer due to the asymmetric osmotic flow of water between the two layers. As is known, specific interactions between ions, polymers, and internal water molecules will modulate the osmotic response depending on the kind of salt and the molecular structure of polymers [28]. In polyacrylic acid-based hydrogels, a coordination bond can be formed between the carboxylic acid and trivalent metal ion that will lead to an AAc polymer chain with a high crosslinking density [29,30,31,32,33,34], resulting in the shrinkage of AAc-based hydrogel networks. Therefore, it is expected that a metal ion and thermal-responsive hydrogel actuator can be achieved due to the asymmetric osmotic flow of water between the two hydrogel layers under different stimuli, with a semi-interpenetrating network interface through the in situ polymerization of the AAc-based pregel layer on the prepared NIPAM hydrogel layer.

Herein, a bilayer hydrogel, with a NIPAM layer hydrogel serving as a thermally responsive layer and an AAc/AAm layer hydrogel serving as the second stimuli-responsive layer, was prepared and investigated. Thermally driven self-deformation behavior was observed due to the thermal-responsive NIPAM layer. The swelling behavior and shrinkage ratio of the two layers of hydrogels immersed in different metal ion aqueous solutions was investigated. The metal ion-responsive shape-morphing behavior, induced by an asymmetric osmotic flow of water between the two hydrogel layers (caused by specific interactions between ions, polymers, and internal water molecules), was investigated in this work. To the best of our knowledge, this is the first study to investigate different metal ion-responsive bilayer hydrogel actuators affected by the shrinkage ratio in the two layers of the hydrogels immersed in different metal ion aqueous solutions. Moreover, the cost of the bilayer actuator was very low, which indicated that the bilayer actuator would be suitable as a controllable switch or smart device for one-time use.

## 2. Experimental

### 2.1. Materials

Acrylic acid (AAc), acrylamide (AAm), and N-isopropyl acrylamide (NIPAM) were acquired from Shanghai Macklin Biochemical Co., Ltd. (Shanghai, China) 2,2-dimethoxy-2-phenylacetophenone (DMPA) (Acros, Waltham, MA, USA) was dissolved in 1-vinyl-2-pyrrolidone (VP) (Aladdin, USA) and used as a photoinitiator. N,N′-methylenebisacrylamide (BIS), N,N,N′,N′-tetraethylethylenediamine (TMEDA), ammonium persulfate, NaCl, CaCl_2_, NiCl_2_∙6H_2_O, AlCl_3_∙6H_2_O, FeCl_3_∙6H_2_O, CrCl_3_∙6H_2_O, and CeCl_3_∙7H_2_O were supplied by Aladdin (Shanghai, China). Rhodamine B was purchased from MYM Biotechnology Co., Ltd (Beijing, China). All other reagents were of analytical grade and used as received.

### 2.2. Preparation of Bilayer Hydrogels (See Figure 1)

Stoichiometric amounts of AAc, AAm, and N,N′-methylenebisacrylamide (BIS) (1‰ wt of AAm and AAc), and 0.1 g/mL 2,2-dimethoxy-2-phenylacetophenone solution (DMPA) (1‰ wt of AAm and AAc) were dissolved in 2 mL of deionized water with a small amount of rhodamine B, which served as a color indicator, to prepare the AAc/AAm pregel. Meanwhile, 1.0 g of N-isopropyl acrylamide (NIPAM), 0.02 g/mL of BIS aqueous solution, 20 mg of ammonium persulphate, and 10 μL of N,N,N′,N′-tetramethylethylenediamine (TMEDA) were added to 4.5 mL of deionized water and stirred until completely dissolved to prepare a NIPAM pregel. Then, the NIPAM solution was cast into rectangular glass molds (100 mm × 100 mm × 3 mm) attached to a silicone plate spacer (dimensional space for casted samples of 80 mm × 80 mm × 1 mm). After covering the molds with another piece of glass, NIPAM hydrogels were formed after a 6 h reaction. Then, a second silicone plate spacer was placed above the first silicone plate spacer after the upper glass plate was lifted. The AAc/AAm pregel solution was cast onto NIPAM hydrogels and exposed to a UV lamp for 2 h after covering the upper glass plate. Finally, the bilayer hydrogel was obtained by thoroughly rinsing with a large amount of water after extraction from the mold.

**Figure 1 polymers-14-04019-f001:**
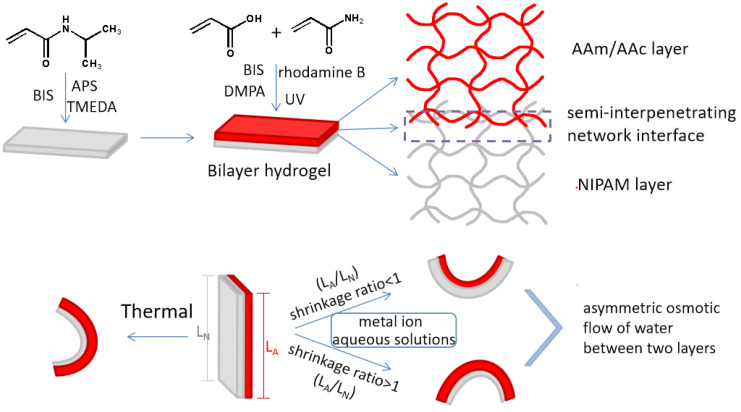
Preparation of the bilayer hydrogel actuator and illustration of the shape-morphing behavior activated by heat or metal ions.

### 2.3. Swelling Behavior

The swelling behavior of the hydrogels [35,36] was measured in deionized water or metal ion solutions. Vacuum-dried samples were carefully weighed prior to immersion in deionized water, 0.6 mol/L NaCl aqueous solution, 0.6 mol/L CaCl_2_ aqueous solution, 0.6 mol/L NiCl_2_ aqueous solution, 0.6 mol/L AlCl_3_ aqueous solution, 0.6 mol/L FeCl_3_ aqueous solution, 0.6 mol/L CrCl_3_ aqueous solution, and 0.6 mol/L CeCl_3_ aqueous solution at room temperature. The specimens were periodically removed from the above solutions at room temperature, followed by wiping with tissue paper to remove any water on the surface, and then weighed using a hundred-thousandths microanalytical balance (AUW120D, Japan, 42/120 g, 0.01/0.1 mg) immediately. The swelling ratio at different times was calculated according to Equation (1):
(1)$$\mathrm{Swelling}\mathrm{ratio}=\frac{W_{\mathrm{st}}}{W_{\mathrm{dry}}}$$
where *W_st_* and *W_dry_* represent the mass of the hydrogel at different times and the initial dry state, respectively. Each swelling data point is expressed as the average of five measurements.

### 2.4. SEM Analysis

The morphologies of the fracture surface of the freeze-dried bilayer hydrogel in a swollen state sprayed with Au−Pd alloy were investigated using an Ultra55 field-emission scanning electron microscope (SEM; Zeiss, Oberkochen, Germany).

### 2.5. Metal Ion and Thermal-Driven Shape-Morphing Behavior

In order to study the bending direction of the hydrogel actuator, rhodamine B (red dye) was used to stain the AAm/AAc layer hydrogel. The double-layer hydrogel, AAm/AAc hydrogel, and NIPAM hydrogel, all in swelling equilibrium state in water, were cut into 30 mm × 4 mm (length × width) strips and then soaked in 0.6 mol/L NaCl aqueous solution, 0.6 mol/L CaCl_2_ aqueous solution, 0.6 mol/L NiCl_2_ aqueous solution, 0.6 mol/L AlCl_3_ aqueous solution, 0.6 mol/L FeCl_3_ aqueous solution, 0.6 mol/L CrCl_3_ aqueous solution, and 0.6 mol/L CeCl_3_ aqueous solution. The self-deformation behavior of these hydrogels was recorded using a camera. As a control, the hydrogel in the swelling equilibrium state was immersed in deionized water to perform the same steps following the above process. In order to better demonstrate the self-deformation behavior, we simulated the process of petal closure through the driving action of metal ions. Under swelling equilibrium conditions, the bilayer hydrogel was cut into four-leaf clover shapes, and then immersed in 0.6 mol/L AlCl_3_ and 0.6 mol/L FeCl_3_ solutions. The specific experimental conditions were the same as those above.
Shrinkage ratio = L_A_/L_N_,
where L_A_ and L_N_ are the lengths of the AAm/AAc hydrogel and NIPAM hydrogel after being immersed in the metal ion solutions for 12 h. The shrinkage rate reflects the bending direction of the hydrogel actuator.

The thermally driven self-deformation behaviors of hydrogels were studied at 60 °C and from room temperature (28 °C) at a heating rate of 4 °C/min. The self-deformation behavior of these hydrogels was recorded using a camera. The size of the hydrogel spline was the same as that of the ion-driven hydrogel spline.

### 2.6. Mechanical Measurements

Cylindrical hydrogel samples in swelling equilibrium state (diameter 12 ± 0.2 mm, thickness 6.5 ± 0.2 mm) were used to perform the compression experiments (BL-GRW005 K electronic universal test, Zwick/Roell, Ulm, Germany) at a constant test speed of 4 mm/min. A rectangular hydrogel sample swelling equilibrium state of 3.0 mm wide × 30 mm long × 1 mm thick was used to perform the tensile test at a tensile speed of 50 mm/min. All mechanical tests were carried out at room temperature. At least three samples were repeated for each test condition, and the average of the three results within the error range was taken.

## 3. Results and Discussion

### 3.1. Swelling Behaviors of Bilayer Hydrogel, AAm/Aac Layer Hydrogel, and NIPAM Layer Hydrogel

The swelling ratios of the bilayer hydrogel, AAm/Aac layer hydrogel, and NIPAM layer hydrogel at different times in deionized water, 0.6 mol/L NaCl aqueous solution, 0.6 mol/L CaCl_2_ aqueous solution, 0.6 mol/L NiCl_2_ aqueous solution, 0.6 mol/L AlCl_3_ aqueous solution, 0.6 mol/L FeCl_3_ aqueous solution, 0.6 mol/L CrCl_3_ aqueous solution, and 0.6 mol/L CeCl_3_ aqueous solution were investigated, as shown in Figure 2. Compared with the swelling ratios of the hydrogels in deionized water, the swelling ratio decreased significantly in the monovalent metal ion solution, the divalent metal ion solution, and the trivalent metal ion solution. The swelling volume difference of hydrogels was attributed to the difference in osmolarity between the inside and outside of the hydrogels. The osmotic equilibrium would be reached by the ion diffusion and the osmolarity-driven migration of water when the hydrogels were immersed in the metal ion solutions. Therefore, the equilibrium swelling ratio of both the AAm/Aac layer hydrogel and the NIPAM layer hydrogel in metal ion aqueous solutions decreased significantly compared with that in deionized water due to the effect of osmotic pressure [28,37,38,39]. Figure 2a shows the swelling ratio of the AAm/Aac layer hydrogel. The water absorption ratio of the AAm/Aac layer hydrogels soaked in different metal ion solutions increased significantly for the first 4 h, and achieved stable values after 12 h. Owing to the osmolarity difference between the inside and outside of the hydrogels, the swelling ratios of the AAm/Aac layer hydrogel in the monovalent metal ion and divalent metal ion solutions were lower than that in deionized water. It has been reported [40] that NaCl, one type of Kosmotropic salt, could promote the formation of a hydrogen-bonding network of water molecules around the AAm/Aac copolymer chains, resulting in a decreased entropy of mixing and an increased UCST of the polymer, and finally decreases the solubility of the polymer in water. Therefore, the swelling ratio of the AAm/AAc layer hydrogel in monovalent metal ion NaCl solution was lower than that in deionized water owing to the synergistic effect of the formation of a H-bonded network and the osmotic pressure difference achieved by salts. The carboxylic acid groups on the AAm/AAc polymer chains could be protonated by acidic solutions, thereby facilitating the formation of a H-bond network in the AAm/AAc layer hydrogel [40]. Divalent metal ion Ca^2+^ and Ni^2+^ solutions are usually weakly acidic, so hydrogen bond networks were more easily formed by the protonation of carboxylic acid groups on the AAm/AAc polymer chains, leading to a decrease in the swelling ratio. Thus, the swelling ratio of the AAm/AAc hydrogel in divalent metal ion solutions was lower than that in the monovalent metal ion NaCl solution. Additionally, the swelling ratio of the AAm/AAc layer hydrogel in the trivalent metal ion solution was lowest among that in deionized water and monovalent metal ion or divalent metal ion solutions, as the synergistic effect of the formation of a H-bonded network achieved by protonation, the osmotic pressure difference, and the coordination of carboxylic acid and trivalent metal ions led to a high crosslinking density of AAm/AAc copolymer chains. The swelling ratio of the NIPAM layer hydrogel is shown in Figure 2b. The water absorption of the NIPAM layer hydrogel after soaking in different metal ion solutions increased significantly within the first 8 h, and reached a stable value after 12 h. The swelling ratio of the NIPAM layer hydrogel showed no obvious difference among the monovalent metal ion solution, divalent metal ion solution, or trivalent metal ion solution, with a slightly higher value in the monovalent metal ion solution, while it decreased significantly compared with that in deionized water. It has been reported [41] that the LCST of PNIPAM would be decreased in trivalent, divalent, and monovalent cation solutions. Although the salting-out effects of different metal ions followed a different order. with the trivalent salt being the most salted out and the monovalent salt solution being the least salted out, the LCST of PNIPAM would be lower than 25 °C when the concentration of cations exceeds 0.2 M. In this work, the concentration of all of the metal ion solutions was 0.6 M, in which the NIPAM polymer chains would undergo phase transitions owing to the decreased LCST lower than 25 °C. From the research publicated by Paul S. Cremer et al. [42], it is known that the ability of a particular anion to lower the LCST of PNIPAM generally followed the Hofmeister series. The LCST would change with the anion concentration and identity. In this work, the anions we used were all Cl^−^. Although the concentration of the anion Cl^−^ doubled or tripled in the divalent metal ion and trivalent metal ion solutions, the decrease in LCST caused by the increase in the Cl^−^ concentration was far lower than the decrease in LCST caused by the increase in the metal cation concentration [41,42]. Therefore, the swelling ratio of the NIPAM layer hydrogel changed as shown in Figure 2b due to the phase transitions caused by the lower LCST of PNIPAM polymer chains affected by the synergistic effect of the concentration and identity of cations and anions. Figure 2c shows the swelling ratio of the bilayer hydrogel. The swelling ratio of the AAm/AAc layer hydrogel in the metal ion solution was greater than that of the NIPAM layer hydrogel. The swelling ratio of the AAm/AAc layer hydrogel in the metal ion solution varied greatly, while the swelling ratio of the NIPAM layer hydrogel in the metal ion solution did not greatly differ; therefore, the swelling ratio of the bilayer hydrogel in the metal ion solution mainly depended on the swelling ratio of the AAm/AAc layer hydrogel.

### 3.2. Metal Ion-Driven Self-Deformation Behavior

Before testing, the cumulative release of Rhodamine B was determined, as shown in Figure 3a. It could be seen that less than 40% of the Rhodamine B was released from the hydrogel and the color of the hydrogel stained with Rhodamine B was still red for seven days, suggesting that most of the rhodamine was still present in the hydrogel and could act as an effective dye. It has been reported [43,44,45] that hydrogels can effectively adsorb cationic dyes in the presence of carboxyl groups. Thus, Rhodamine B acted as an effective dye during the following test to better illustrate the shape-morphing behavior of the bilayer hydrogel. SEM was used to investigate the interfacial structure of the bilayer hydrogel, as shown in Figure 3b,c. A strongly bonded interface with some interpenetrating structures could be seen in the SEM images, and the thickness of the bonded interface (in the red box) calculated from the SEM images was approximately 6 μm. In Figure 3d, we tried to peel off the bilayer hydrogels in a swollen state, and in any case, the bilayer hydrogel could not be peeled off to obtain a complete NIPAM hydrogel layer. After destroying the NIPAM hydrogel layer, the interface of the bilayer hydrogel showed a rough state. These findings indicate that a bilayer hydrogel with a strong adhesive interface was successfully prepared in this work.

As is known, metal ion solutions are usually acidic. In order to be convenient and reliable for practical applications, the ion solution during the test maintained its pH value without adjustment. The bilayer hydrogel bent toward the AAm/AAc layer when immersed in a Fe^3+^ aqueous solution, while immersion in 0.6 mol/L NaCl, 0.6 mol/L CaCl_2_, 0.6 mol/L NiCl_2_, 0.6 mol/L AlCl_3_, 0.6 mol/L CrCl_3_, or 0.6 mol/L CeCl_3_ aqueous solution caused the bilayer hydrogel actuators to bend toward the NIPAM layer (See Figure 4). The bending angle of the bilayer hydrogel actuators was almost 180° after 12 h of immersion in the metal ion solution, indicating that the bilayer hydrogel actuator had a good metal ion-responsive ability. Figure 5 shows the length-shrinkage behavior of the AAm/AAc layer hydrogel and NIPAM layer hydrogel after soaking in the metal ion solution for 12 h. The shrinkage behavior of the hydrogel, which was caused by the synergistic effect of the osmolarity difference between the inside and outside of the hydrogels and the difference in the interaction between the ions and the hydrogel molecular chain [13,22,29], was similar to the de-swelling behavior. When the hydrogel in a swollen equilibrium state was soaked in a metal ion solution, a different shrinkage behavior occurred, which was closely related to the swelling behavior of the hydrogels (see Figure 2). The swelling ratios of the AAm/AAc layer hydrogel and NIPAM layer hydrogel were highest in deionized water, while they decreased in metal ion solutions. Thus, the length of the swollen hydrogel was shorter when immersed in metal ion solutions. Owing to the osmolarity difference between the inside and outside of the hydrogels, the AAm/AAc layer hydrogel shrank to a certain extent when immersed in the metal ion aqueous solution. While the AAm/AAc layer hydrogel shrank the most in a trivalent metal ion solution, the synergistic effect of the osmotic pressure and the coordination of carboxylic acid and trivalent metal ions led to a high crosslinking density of AAm/AAc copolymer chain [29,30]. Regarding the synergistic effect of the osmolarity difference between the inside and outside of the hydrogels and the interaction difference between the ions and the NIPAM polymer chain, the NIPAM layer hydrogel shrank the least when immersed in a Fe^3+^ solution, but showed no obvious shrinkage difference when immersed in other metal ion solutions. Therefore, except for the hydrogel immersed in Fe^3+^, the shrinkage ratio of the two layers of the hydrogels immersed in other metal ion solutions was greater than 1, as shown in Table 1. Finally, the bilayer hydrogel bent toward the AAm/AAc layer when immersed in Fe^3+^ aqueous solution, and toward the NIPAM layer when immersed in the other metal ion aqueous solutions. This indicates that bilayer hydrogel actuators can be used as ion-controllable switches or other smart devices.

In addition, in order to better demonstrate the self-deformation behavior, four-leaf-clover-shaped hydrogel actuators were used to investigate the metal ion-responsive self-deformation behavior when immersed in 0.6 mol/L AlCl_3_ solution and 0.6 mol/L FeCl_3_ solution, respectively (see Figure 6). The process of petal closure was simulated by the shape-morphing behavior of the hydrogel actuator under metal ion stimulation. The four-leaf clover hydrogel actuator bent toward the AAm/AAc layer (red layer) when immersed in the Fe^3+^ solution, and toward the NIPAM layer (white layer) when immersed in the Al^3+^ solution.

### 3.3. Thermally Driven Self-Deformation Behavior

Figure 7 shows the self-deformation behavior of the bilayer hydrogel actuators at a constant temperature of 60 °C or at a heating rate of 4 °C/min starting from room temperature (28 °C). It shows that the bilayer hydrogel actuator exhibited obvious self-deformation behavior after 1 min, and deformed from a long strip to an approximate semicircle after 9 min. In addition, the self-deformation process of the bilayer hydrogel at 60 °C was slightly faster than that of the bilayer hydrogel at a heating rate of 4 °C/min. The lower critical phase-transition temperature (LCST) of the NIPAM layer hydrogel was about 32 °C. At higher temperatures, NIPAM blocks might undergo phase transition, causing the shrinkage of the hydrogel networks. Thus, during the heating process, the volume of the hydrogel of the NIPAM layer decreased, while the volume of the hydrogel of the AAm/AAc layer did not change significantly. Therefore, the bilayer hydrogel actuator bent toward the NIPAM layer.

### 3.4. Mechanical Properties

Figure 8 shows the tensile and compressive stress–strain curves of the bilayer hydrogel, the AAm/AAc layer hydrogel, and the NIPAM layer hydrogel. The NIPAM layer hydrogel had lower tensile and compressive strength, while the AAm/AAc layer hydrogel had better mechanical strength. The tensile strength and compressive strength of the bilayer hydrogel ranked between those of the AAm/AAc layer hydrogel and the NIPAM layer hydrogel, indicating that the mechanical properties of NIPAM hydrogels could be reinforced by AAm/AAc hydrogels.

## 4. Conclusions

In this work, metal ion- and thermal-responsive shape-morphing hydrogel actuators with good mechanical properties were prepared and investigated. Under thermal stimuli, the hydrogel actuators bent toward the NIPAM layer due to the shrinkage of the hydrogel networks caused by the hydrophilic–hydrophobic phase-transition of NIPAM blocks above the LCST. For the synergistic effect of the osmolarity difference between the inside and outside of the hydrogels and the interaction difference between ion and hydrogel polymer chains, the shrinkage ratio of the AAm/AAc layer to NIPAM layer immersed in Fe^3+^ solutions was lower than 1, while the shrinkage ratio of the AAm/AAc layer to the NIPAM layer immersed in NaCl, CaCl_2_, NiCl_2_, AlCl_3_, CrCl_3_, and CeCl_3_ aqueous solutions was higher than 1. Therefore, the bilayer hydrogel actuator bent toward the AAm/AAc layer when immersed in Fe^3+^ aqueous solution, while it bent toward the NIPAM layer when immersed in other metal ion aqueous solutions. This indicates that the bilayer hydrogel actuators could be used as ion or temperature-controllable switches or other smart devices.

## Figures and Tables

**Figure 2 polymers-14-04019-f002:**
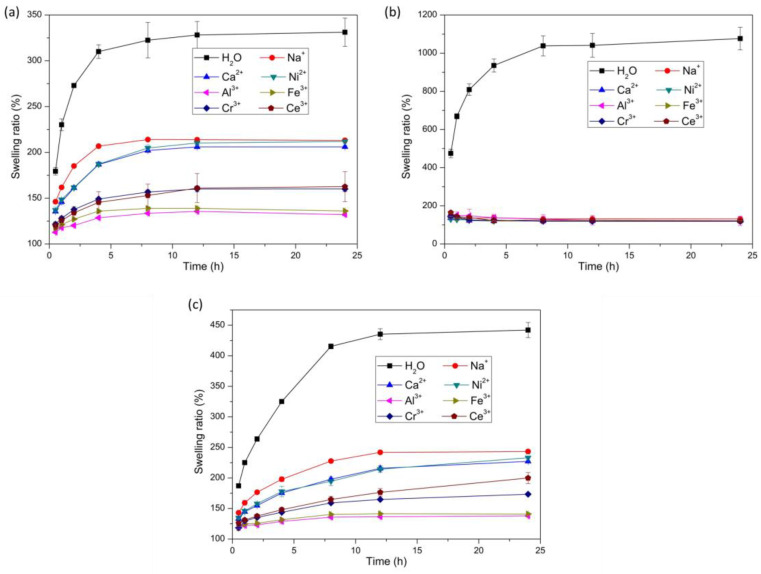
Swelling behavior of AAc/AAm hydrogels (**a**), NIPAM hydrogels (**b**), and bilayer hydrogels (**c**) soaked in deionized water, 0.6 mol/L NaCl solution, 0.6 mol/L CaCl_2_ solution, 0.6 mol/L NiCl_2_ solution, 0.6 mol/L AlCl_3_ solution, 0.6 mol/L FeCl_3_ solution, 0.6 mol/L CrCl_3_ solution, and 0.6 mol/L CeCl_3_ solution. (*n* = 3).

**Figure 3 polymers-14-04019-f003:**
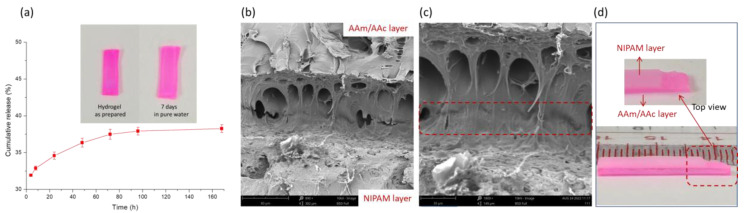
(**a**) Cumulative release of Rhodamine B from AAc/AAm hydrogel (*n* = 3), (**b**) low-magnification and (**c**) high-magnification SEM images; (**d**) photos of the bilayer hydrogels in swollen state.

**Figure 4 polymers-14-04019-f004:**
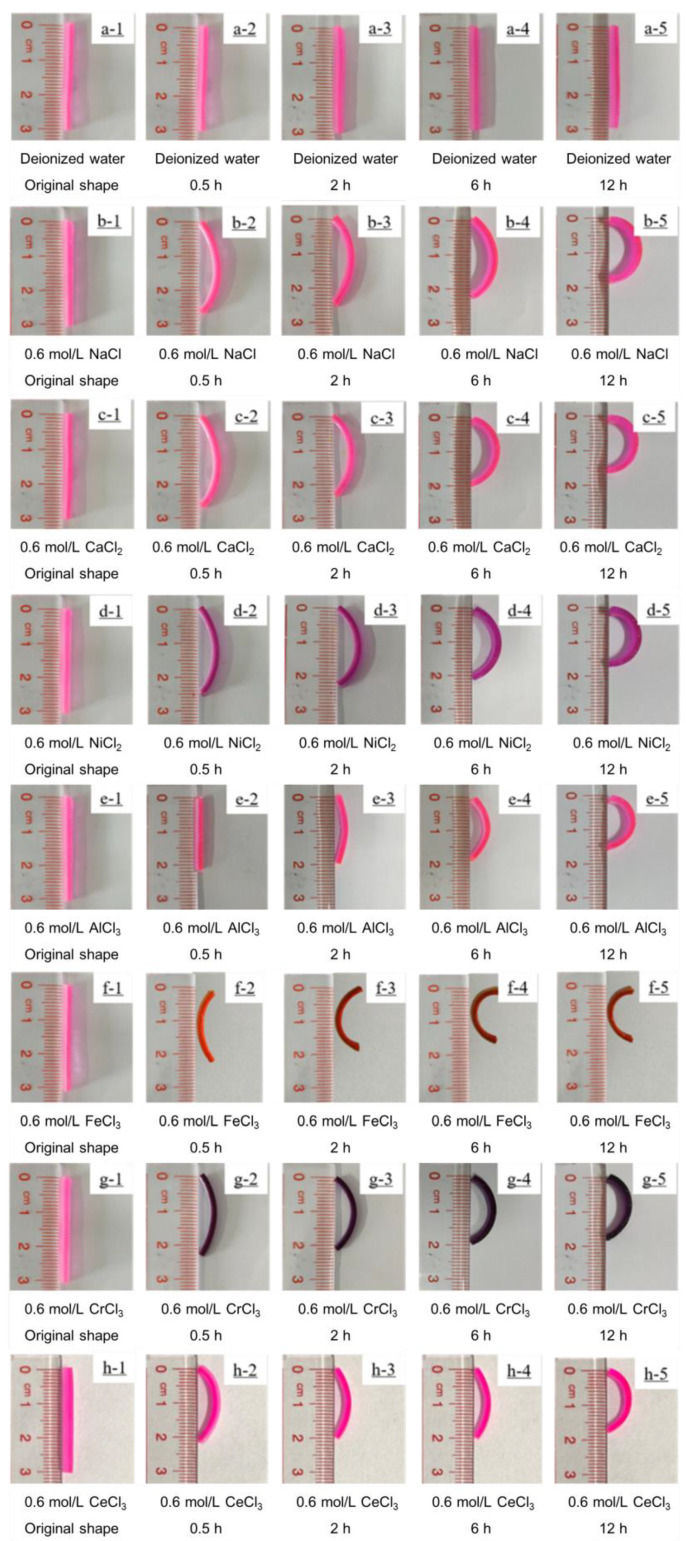
Self-deformation behavior of bilayer hydrogel actuators in 0.6 mol/L NaCl solution (**b1**–**b5**), 0.6 mol/L CaCl_2_ solution (**c1**–**c5**), 0.6 mol/L NiCl_2_ solution (**d1**–**d5**), 0.6 mol/L AlCl_3_ solution (**e1**–**e5**), 0.6 mol/L FeCl_3_ solution (**f1**–**f5**), 0.6 mol/L CrCl_3_ solution (**g1**–**g5**), and 0.6 mol/L CeCl_3_ (**h1**–**h5**). Hydrogel actuators were placed in deionized water (**a1**–**a5**) as a control. In all images, the NIPAM hydrogel layer is on the left and the AAm/AAc hydrogel layer is on the right.

**Figure 5 polymers-14-04019-f005:**
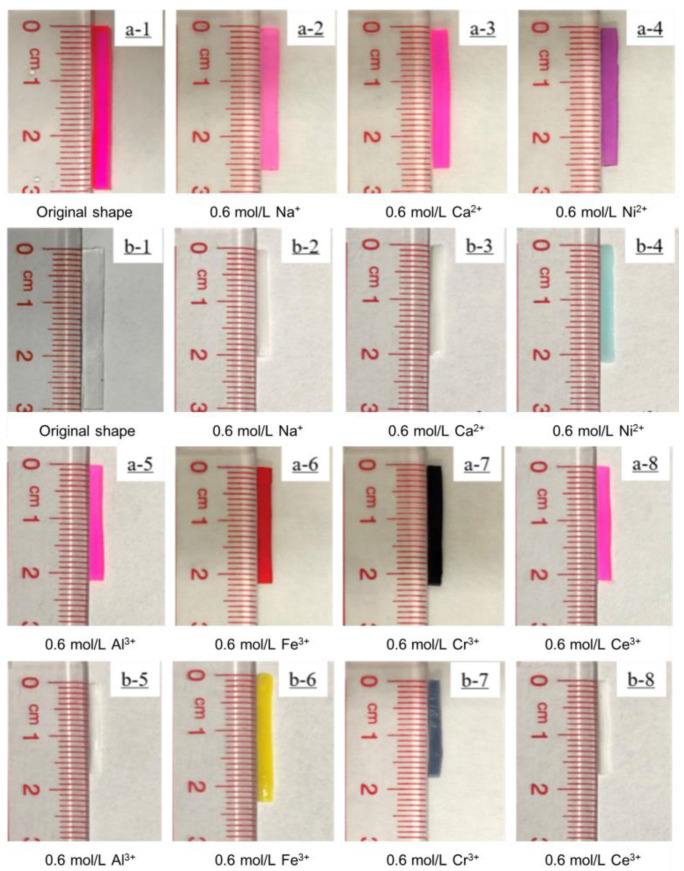
(**a1**–**a8**) AAc/AAm and (**b1**–**b8**) NIPAM hydrogels in 0.6 mol/L NaCl, 0.6 mol/L CaCl_2_, 0.6 mol/L NiCl_2_, 0.6 mol/L AlCl_3_, 0.6 mol/L FeCl_3_, 0.6 mol/L CrCl_3_, and 0.6 mol/L CeCl_3_ solutions after 12 h.

**Figure 6 polymers-14-04019-f006:**
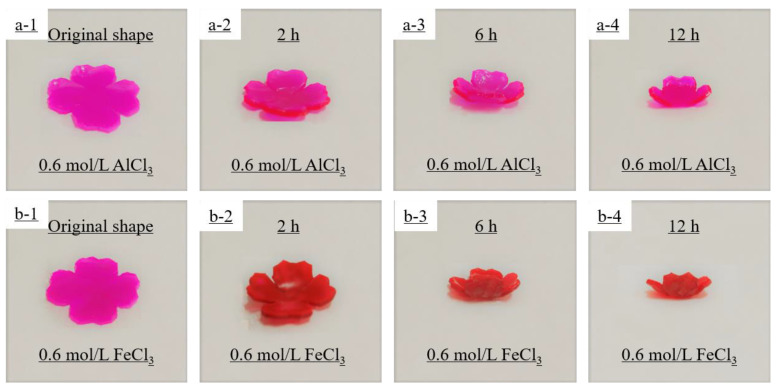
Simulative petal closure process of the four-leaf clover hydrogel actuator when immersed in 0.6 mol/L AlCl_3_ (**a1**–**a4**), with the NIPAM hydrogel layer on top and the AAm/AAc hydrogel layer on the bottom, or 0.6 mol/L FeCl_3_ (**b1**–**b4**), with the AAm/AAc hydrogel layer on top and the NIPAM hydrogel layer on the bottom.

**Figure 7 polymers-14-04019-f007:**
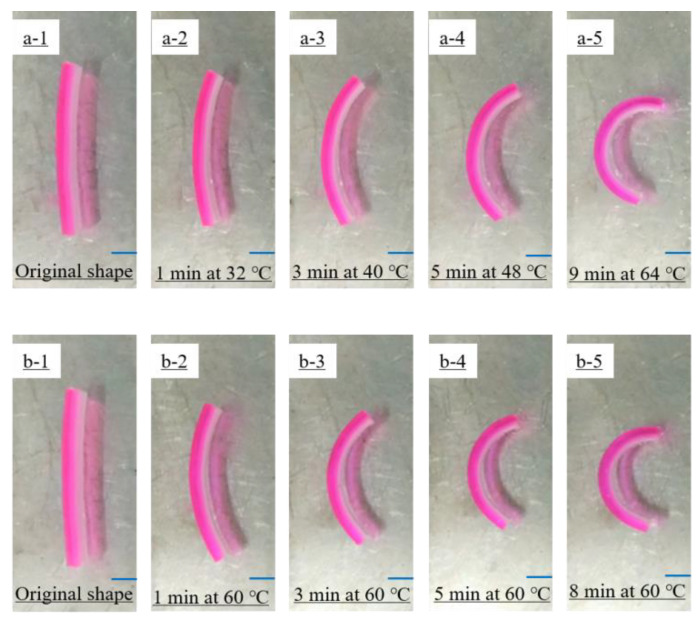
Self-deformation behavior of bilayer hydrogel actuators (**a1**–**a5**) under heating from room temperature (28 °C) at a heating rate of 4 °C/min and (**b1**–**b5**) at a constant temperature of 60 °C. In all images, the AAm/AAc hydrogel layer is on the left and the NIPAM hydrogel layer is on the right.

**Figure 8 polymers-14-04019-f008:**
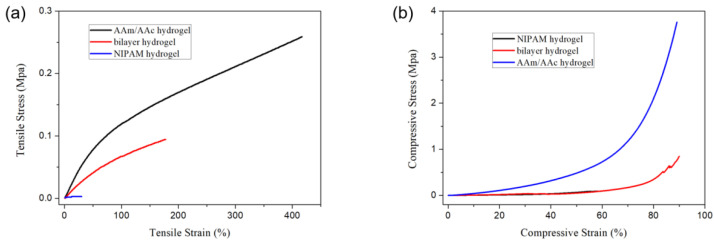
(**a**) Tensile stress−strain curves and (**b**) compressive stress−strain curves of the AAm/AAc hydrogel, NIPAM hydrogel, and bilayer hydrogel at room temperature.

**Table 1 polymers-14-04019-t001:** Shrinkage ratio of the AAm/AAc layer hydrogel to the NIPAM layer hydrogel in different metal ion aqueous solutions, presented as the mean ± SD (*n* = 3).

	AAc/AAm Hydrogel Length (L_A_)/cm	NIPAM Hydrogel Length (L_N_)/cm	L_A_/L_N_
H_2_O	3.00 ± 0.00	3.00 ± 0.00	1.00
0.6 mol/L NaCl aqueous solution	2.60 ± 0.00	2.00 ± 0.06	1.30
0.6 mol/L CaCl_2_ aqueous solution	2.55 ± 0.05	2.00 ± 0.03	1.28
0.6 mol/L NiCl_2_ aqueous solution	2.50 ± 0.10	2.10 ± 0.06	1.19
0.6 mol/L AlCl_3_ aqueous solution	2.15 ± 0.02	1.65 ± 0.00	1.30
0.6 mol/L FeCl_3_ aqueous solution	2.15 ± 0.05	2.30 ± 0.02	0.94
0.6 mol/L CrCl_3_ aqueous solution	2.20 ± 0.05	1.75 ± 0.00	1.26
0.6 mol/L CeCl_3_ aqueous solution	2.10 ± 0.00	1.70 ± 0.00	1.24

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
