# Peer review of "A Metal Ion and Thermal-Responsive Bilayer Hydrogel Actuator Achieved by the Asymmetric Osmotic Flow of Water between Two Layers under Stimuli"

_polymers, 2022, doi:10.3390/polym14194019_

Round 1

Reviewer 1 Report (Previous Reviewer 1)

In this manuscript, the authors present the synthesis and characterization of bilayer hydrogels composed of acrylic acid, acrylamide and N-isopropyl acrylamide. In this resubmission, the authors have addressed some of my concerns regarding the rhodamine B leaching and performed additional SEM imaging as well as quantification of the swelling ratio data. While I appreciate the authors efforts, the newly added sections seem to have some issues, which I will point out below:

- The estimation of molar mass between entanglement (Mc) and the corresponding mesh sizes given in page 4 (refs 37-39) do not make sense to me. I cannot access Ref. 37 myself so cannot verify the claims here, but how did you determine the solid density? The hexane displacement-based density measurement is not described anywhere in the experimental procedure. How were the Flory-Huggins parameter determined for these polymers? Finally, the crosslinking density has a unit (mol/cm^3 or related), so it doesnft make sense to define it as 1 as the paper claims. What are these numbers? If I remember correctly, the original mesh size paper for PVA gels (Peppas, 1989?) had a similar equation to what is shown here, but this paper was for swollen crosslinked gels (and thus is not relevant for NIPAm gels that are shrunk), and they started with a non-crosslinked polymer solution with a known Mn. 

- The lack of error bars in all of the numbers and graphs shown in this manuscript worries me. How many measurements were performed, and what are the errors?

- The methods for preparing SEM samples are not described. Furthermore, SEM is not an ideal method for determining the IPN nature of such molecules. Having covalently fluorophore-labeled gels in one or both of the networks, however, would have opened an avenue to performing fluorescence microscopy on thin crosssections of the gels.

- The PNIPAAm LCST is known to be affected by both cations and anions, with the effects observed governed by the Hoffmeister and reverse Hoffmeister series (https://pubs.acs.org/doi/10.1021/ja0546424, https://pubs.acs.org/doi/10.1021/am9007006). The PAAm/PAAc swelling would be governed by pH and salt concentration in a typical weak polyelectrolyte behavior, although it was recently reported to show UCST in acidic solutions similar to some of the authorsf environment due to the disruption of hydrogen-bonding between carboxylic acids at higher temperatures (https://pubs.acs.org/doi/10.1021/acs.macromol.1c00952). The authors should modify the discussion of swelling behavior comparisons based on this body of knowledge.

In light of these new points raised, I would again suggest a major revision of the paper.

Author Response

In this manuscript, the authors present the synthesis and characterization of bilayer hydrogels composed of acrylic acid, acrylamide and N-isopropyl acrylamide. In this resubmission, the authors have addressed some of my concerns regarding the rhodamine B leaching and performed additional SEM imaging as well as quantification of the swelling ratio data. While I appreciate the authors efforts, the newly added sections seem to have some issues, which I will point out below:

We have revised our manuscript as best as we could, according to the reviewers' comments and the editor’ requirements. The detailed responses to comments are listed below. We really hope our revised manuscript could be suitable for the publication in Polymers”.

- The estimation of molar mass between entanglement (Mc) and the corresponding mesh sizes given in page 4 (refs 37-39) do not make sense to me. I cannot access Ref. 37 myself so cannot verify the claims here, but how did you determine the solid density? The hexane displacement-based density measurement is not described anywhere in the experimental procedure. How were the Flory-Huggins parameter determined for these polymers? Finally, the crosslinking density has a unit (mol/cm^3 or related), so it does not make sense to define it as 1 as the paper claims. What are these numbers? If I remember correctly, the original mesh size paper for PVA gels (Peppas, 1989?) had a similar equation to what is shown here, but this paper was for swollen crosslinked gels (and thus is not relevant for NIPAm gels that are shrunk), and they started with a non-crosslinked polymer solution with a known Mn. 

Response:Thank you very much for improving our manuscript; In the previous revision, another reviewer proposed to supply the calculation of crosslinking density referred to those literatures. However, I re-researched these literatures carefully, and combined with your comments, this calculation method is indeed inaccurate and not relevant for hydrogels that are shrunk, therefore, we removed the section on the calculation of crosslink density, in order to ensure the rigorous correctness of this work. And we firmly believe deleting this part of the calculation does not affect the integrity of this paper, and it is still suitable for publication. We really hope the reviewer can agree with this part we had modified in our revised manuscript. Thanks.

- The lack of error bars in all of the numbers and graphs shown in this manuscript worries me. How many measurements were performed, and what are the errors?

Response:Thank you very much for improving our manuscript; Number of tests and error bars had been added in the relevant part of our revised manuscript.

Figure 2 Swelling behavior of AAc/AAm hydrogels (a), NIPAM hydrogels (b), and bilayer hydrogels (c) soaked in deionized water, 0.6 mol/L NaCl solution, 0.6 mol/L CaCl2 solution, 0.6 mol/L NiCl2 solution, 0.6 mol/L AlCl3 solution, 0.6 mol/L FeCl3 solution, 0.6 mol/L CrCl3 solution, and 0.6 mol/L CeCl3 solution. (n=3)

Figure 3 (a) the cumulative release of Rhodamine B from AAc/AAm hydrogel (n=3), (b) Low and (c) high magnification SEM images; (d) Photos of the bilayer hydrogels in swollen state.

Table 1 Shrinkage ratio of AAm/AAc layer hydrogel to NIPAM layer hydrogel in different metal ion aqueous solution, presented as mean ± SD (n = 3).

AAc/AAm hydrogel

length (LA)/ cm

NIPAM hydrogel length (LN)/ cm

LA/LN

H2O

3.00±0

3.00±0

1.000

0.6 mol/L NaCl aqueous solution

2.60±0

2.00±0.067

1.300

0.6 mol/L CaCl2 aqueous solution

2.55±0.05

2.00±0.033

1.275

0.6 mol/L NiCl2 aqueous solution

2.50±0.10

2.10±0.067

1.190

0.6 mol/L AlCl3 aqueous solution

2.15±0.025

1.65±0

1.303

0.6 mol/L FeCl3 aqueous solution

2.15±0.05

2.30±0.025

0.935

0.6 mol/L CrCl3 aqueous solution

2.20±0.05

1.75±0

1.257

0.6 mol/L CeCl3 aqueous solution

2.10±0

1.70±0

1.235

- The methods for preparing SEM samples are not described. Furthermore, SEM is not an ideal method for determining the IPN nature of such molecules. Having covalently fluorophore-labeled gels in one or both of the networks, however, would have opened an avenue to performing fluorescence microscopy on thin crosssections of the gels.

Response:Thank you very much for improving our manuscript; The methods for preparing SEM samples had been added in our revised manuscript. Fluorescence is usually used to covalently label macromolecules, but it is difficult to achieve the labeling of gel network and vinyl monomers. In this paper, we tried to covalently graft FITC onto acrylamide, but the grafting rate was low, analysis purification is difficult and this can hinder the polymerization of the hydrogel. And we tried to add FITC label to the AAm/AAc gel prepolymer solution to label the networks, the small molecule FITC will diffuse into the NIPAM layer, as shown in the figure below, although a clear interface can be seen, the interpenetrating network structure of the interface still cannot been determined. We checked some literature, and also consulted many experts; it is still difficult for us to deeply characterize the interface interpenetrating network. However, interpenetrating network on the interface was mainly used to improve interfacial adhesion performance in this work. And the interface does not appear to be peeled off during the whole test process, metal ion and thermal responsive bilayer hydrogel actuator was successfully achieved in our work. In Figure 3d, we tried to peel off the bilayer hydrogels in swollen state, and in any case, the bilayer hydrogel could not be peeled off to obtain a complete NIPAM hydrogel layer. After destroying the NIPAM hydrogel layer, the interface of the bilayer hydrogel showed a rough state. These indicated that bilayer hydrogel with strong adhesive interface was successfully prepared in this work. We had revised our manuscript as best as we could, according to your useful suggestions. We really hope our revised manuscript could be suitable for the publication in “Polymers”.

- The PNIPAAm LCST is known to be affected by both cations and anions, with the effects observed governed by the Hoffmeister and reverse Hoffmeister series (https://pubs.acs.org/doi/10.1021/ja0546424, https://pubs.acs.org/doi/10.1021/am9007006). The PAAm/PAAc swelling would be governed by pH and salt concentration in a typical weak polyelectrolyte behavior, although it was recently reported to show UCST in acidic solutions similar to some of the authorsf environment due to the disruption of hydrogen-bonding between carboxylic acids at higher temperatures (https://pubs.acs.org/doi/10.1021/acs.macromol.1c00952). The authors should modify the discussion of swelling behavior comparisons based on this body of knowledge.

Response:Thank you very much for improving our manuscript; the discussion of swelling behavior comparisons based on this body of knowledge had been modified in our revised manuscript. It was reported 1 that NaCl, one kind of Kosmotropic salts, could promote the formation of a hydrogen-bonding network of water molecules around the AAm/AAc copolymer chains, resulting in a decreased entropy of mixing and an increased UCST of the polymer, and finally would decrease the solubility of the polymer in water. Therefore, the swelling ratio of the AAm/AAc layer hydrogel in monovalent metal ion NaCl solution is lower than that in deionized water, owing to the synergistic effect of formation of H-bonded network and the osmotic pressure difference achieved by salts. The carboxylic acid groups on the AAm/AAc polymer chains could be protonated by acidic solutions, thereby facilitating the formation of H-bond network in the AAm/AAc layer hydrogel. 1 Divalent metal ion Ca2+ and Ni2+ solutions were usually weakly acidic, so hydrogen bond networks were more easily formed by protonation of carboxylic acid groups on the AAm/AAc polymer chains, leading to the decrease of swelling ratio. Thus, the swelling ratio of AAm/AAc hydrogel in divalent metal ion solutions was lower than that in monovalent metal ion NaCl solution. And the swelling ratio of the AAm/AAc layer hydrogel in trivalent metal ion solution ranked least among that in deionized water and monovalent metal ion or divalent metal ion solution, for the synergistic effect of formation of H-bonded network achieved by protonation, the osmotic pressure difference and the coordination of carboxylic acid and trivalent metal ion which led to a high crosslinking density of AAm/AAc copolymer chains. The swelling ratio of the NIPAM layer hydrogel was shown in Figure 2b. The water absorption of the NIPAM layer hydrogel after soaking in different metal ion solutions increased significantly within the first 8 h, and reached a stable value after 12 h. The swelling ratio of the NIPAM layer hydrogel showed no obvious difference in the monovalent metal ion solution, divalent metal ion solution, or trivalent metal ion solution with slightly higher in the monovalent metal ion solution, while it decreased significantly compared with that in deionized water. It was reported 2 that LCST of PNIPAM would be decreased in tri-, divalent and monovalent cation solutions. Although the salting-out effects of different metal ions followed a different order with the trivalent salt being the most salted out, and the monovalent salt solution being the least salted out, the LCST of PNIPAM would be lower than 25 ℃ when the concentration of cations exceeds 0.2 M. In this work, the concentration of all the metal ion solutions was 0.6 M, in which the NIPAM polymer chains would undergo phase transitions owing to decreased LCST lower than 25 ℃. From the research publicated by Paul S. Cremer et al 3, it could be known that the ability of a particular anion to lower the LCST of PNIPAM generally followed the Hofmeister series. The LCST would be changed with anion concentration and identity. In this work, the anions we use were all Cl-. Although the concentration of anion Cl- increased to double or triple in divalent metal ion and trivalent metal ion solution, the decrease in LCST caused by the increase in Cl- concentration is far less than the decrease in LCST caused by the increase in metal cation concentration. 2-3 Therefore, the swelling ratio of the NIPAM layer hydrogel showed the change as shown in Figure 2b, owing to the phase transitions caused by lower LCST of PNIPAM polymer chains affected by the synergistic effect of concentration and identity of cations and anions.

In light of these new points raised, I would again suggest a major revision of the paper.

We have revised our manuscript as best as we could, according to your comments. We really hope our revised manuscript could be suitable for the publication in Polymers”. Thanks

Thank you very much for all your help and looking forward to hearing from you soon.

Best regards

Yours sincerely

Huilong Guo

Institute Of Biological And Medical Engineering, Guangdong Academy of Sciences Contact email: huilongguo@126.com

REFERENCES

  1. Beaudoin, G.; Lasri, A.; Zhao, C.; Liberelle, B.; De Crescenzo, G.; Zhu, X.-X., Making Hydrophilic Polymers Thermoresponsive: The Upper Critical Solution Temperature of Copolymers of Acrylamide and Acrylic Acid. Macromolecules 2021, 54 (17), 7963-7969.
  2. Fu, H.; Hong, X.; Wan, A.; Batteas, J. D.; Bergbreiter, D. E., Parallel Effects of Cations on PNIPAM Graft Wettability and PNIPAM Solubility. Acs Appl Mater Inter 2010, 2 (2), 452-458.
  3. Zhang, Y.; Furyk, S.; Bergbreiter, D. E.; Cremer, P. S., Specific Ion Effects on the Water Solubility of Macromolecules:  PNIPAM and the Hofmeister Series. Journal of the American Chemical Society 2005, 127 (41), 14505-14510.

Reviewer 2 Report (New Reviewer)

This paper is about a bilayer shape-morphing actuator responsive to metal ions or temperature. It is claimed that the designed bilayer hydrogel actuator can be potentially used as controllable switches or other smart devices. There are some suggestions to further enhance the manuscript.

1.      The novelty of this manuscript is that the bilayer actuator can bend when immersed in metal ion solutions or undergoing temperature change due to the synergistic effect of osmolarity difference and interaction between ions and polymer chains.  However, the interaction between ions and polymer chains is not explained clearly.

a.      The author only specified the interaction between AAc/AAm and ion, and didn’t target the detailed interaction between NIPAM and ions.

b.      Need to explain Figure 2 more. Why swelling ratios of NIPAM at equilibrium showed almost no difference among variable metal ions, whereas those varied for AAc/AAm hydrogel?

c.      Also need an explanation in Figure 2a, what caused the different swelling ratios for AAc/AAm hydrogel among metal ions. Due to the ionic size or other reasons? Why this phenomenon only seen in AAc/AAm not NIPAM?

2.      From the bilayer preparation method, the NIPAM layer was prepared first, and AAc/AAm pre gel was cast onto NIPAM afterwards, following with the UV cure.

a.      Figure 3b and 3c, please mark the name of each layer and the thickness of the interpenetrated interface. Please briefly introduce how you prepared the SEM sample, freeze-dry under swollen equilibrium state or under as-prepared state.

b.      The bonding force between two layers has to be proved, especially in the swollen equilibrium state.

c.      Please mention what solvent was used.  ‘The double-layer hydrogel, AAm/AAc hydrogel and NIPAM hydrogel all in swelling equilibrium state were cut into 30 mm × 4 mm (length × width) strips…’

3.      In Figure 5, which is AAc/AAm hydrogel and which is NIPAM? Please include details.

4.      In Figure 6, please draw a simple scheme of the device structure. What is the top layer, and what is the bottom layer?

5.      The bilayer actuator was only demonstrated with one-time use, i.e. bending when soaked in metal ions. If wanted to be used as a controllable switch or smart device, from the bending to a non-bending state needs to be proved, it would add novelty to realize a reversible behavior. 

Author Response

This paper is about a bilayer shape-morphing actuator responsive to metal ions or temperature. It is claimed that the designed bilayer hydrogel actuator can be potentially used as controllable switches or other smart devices. There are some suggestions to further enhance the manuscript.

We have revised our manuscript as best as we could, according to the reviewers' comments and the editor’ requirements. The detailed responses to comments are listed below. We really hope our revised manuscript could be suitable for the publication in Polymers”.

  1. The novelty of this manuscript is that the bilayer actuator can bend when immersed in metal ion solutions or undergoing temperature change due to the synergistic effect of osmolarity difference and interaction between ions and polymer chains.  However, the interaction between ions and polymer chains is not explained clearly.

Response:Thank you very much for improving our manuscript; the interaction between ions and polymer chains had been explained clearly in our revised manuscript.

  1. The author only specified the interaction between AAc/AAm and ion, and didn’t target the detailed interaction between NIPAM and ions.

Response:Thank you very much for improving our manuscript; the detailed interaction between NIPAM and ions had been added in our revised manuscript.

  1. Need to explain Figure 2 more. Why swelling ratios of NIPAM at equilibrium showed almost no difference among variable metal ions, whereas those varied for AAc/AAm hydrogel?

Response:Thank you very much for improving our manuscript; the reasons had been added in our revised manuscript.

  1. Also need an explanation in Figure 2a, what caused the different swelling ratios for AAc/AAm hydrogel among metal ions. Due to the ionic size or other reasons? Why this phenomenon only seen in AAc/AAm not NIPAM?

Response:Thank you very much for improving our manuscript; an explanation for this phenomenon had been added in our revised manuscript.

The modified revised manuscript of this part is attended below.

It was reported 1 that NaCl, one kind of Kosmotropic salts, could promote the formation of a hydrogen-bonding network of water molecules around the AAm/AAc copolymer chains, resulting in a decreased entropy of mixing and an increased UCST of the polymer, and finally would decrease the solubility of the polymer in water. Therefore, the swelling ratio of the AAm/AAc layer hydrogel in monovalent metal ion NaCl solution is lower than that in deionized water, owing to the synergistic effect of formation of H-bonded network and the osmotic pressure difference achieved by salts. The carboxylic acid groups on the AAm/AAc polymer chains could be protonated by acidic solutions, thereby facilitating the formation of H-bond network in the AAm/AAc layer hydrogel. 1 Divalent metal ion Ca2+ and Ni2+ solutions were usually weakly acidic, so hydrogen bond networks were more easily formed by protonation of carboxylic acid groups on the AAm/AAc polymer chains, leading to the decrease of swelling ratio. Thus, the swelling ratio of AAm/AAc hydrogel in divalent metal ion solutions was lower than that in monovalent metal ion NaCl solution. And the swelling ratio of the AAm/AAc layer hydrogel in trivalent metal ion solution ranked least among that in deionized water and monovalent metal ion or divalent metal ion solution, for the synergistic effect of formation of H-bonded network achieved by protonation, the osmotic pressure difference and the coordination of carboxylic acid and trivalent metal ion which led to a high crosslinking density of AAm/AAc copolymer chains. The swelling ratio of the NIPAM layer hydrogel was shown in Figure 2b. The water absorption of the NIPAM layer hydrogel after soaking in different metal ion solutions increased significantly within the first 8 h, and reached a stable value after 12 h. The swelling ratio of the NIPAM layer hydrogel showed no obvious difference in the monovalent metal ion solution, divalent metal ion solution, or trivalent metal ion solution with slightly higher in the monovalent metal ion solution, while it decreased significantly compared with that in deionized water. It was reported 2 that LCST of PNIPAM would be decreased in tri-, divalent and monovalent cation solutions. Although the salting-out effects of different metal ions followed a different order with the trivalent salt being the most salted out, and the monovalent salt solution being the least salted out, the LCST of PNIPAM would be lower than 25 ℃ when the concentration of cations exceeds 0.2 M. In this work, the concentration of all the metal ion solutions was 0.6 M, in which the NIPAM polymer chains would undergo phase transitions owing to decreased LCST lower than 25 ℃. From the research publicated by Paul S. Cremer et al 3, it could be known that the ability of a particular anion to lower the LCST of PNIPAM generally followed the Hofmeister series. The LCST would be changed with anion concentration and identity. In this work, the anions we use were all Cl-. Although the concentration of anion Cl- increased to double or triple in divalent metal ion and trivalent metal ion solution, the decrease in LCST caused by the increase in Cl- concentration is far less than the decrease in LCST caused by the increase in metal cation concentration. 2-3 Therefore, the swelling ratio of the NIPAM layer hydrogel showed the change as shown in Figure 2b, owing to the phase transitions caused by lower LCST of PNIPAM polymer chains affected by the synergistic effect of concentration and identity of cations and anions.

  1. From the bilayer preparation method, the NIPAM layer was prepared first, and AAc/AAm pre gel was cast onto NIPAM afterwards, following with the UV cure.
  2. Figure 3b and 3c, please mark the name of each layer and the thickness of the interpenetrated interface. Please briefly introduce how you prepared the SEM sample, freeze-dry under swollen equilibrium state or under as-prepared state.

Response:Thank you very much for improving our manuscript; the name of each layer, the thickness of interpenetrated interface and the method for SEM had been added in our revised manuscript.

Figure 3 (a) the cumulative release of Rhodamine B from AAc/AAm hydrogel (n=3), (b) Low and (c) high magnification SEM images; (d) Photos of the bilayer hydrogels in swollen state.

The thickness of bonded interface (in the red box) calculated from SEM images was approximately 6 μm.

The morphologies of fracture surface of the freeze-dried bilayer hydrogel in swollen state sprayed by Au−Pd alloy were investigated by an Ultra55 field emission scanning electron microscope (SEM; Zeiss, Germany).

  1. The bonding force between two layers has to be proved, especially in the swollen equilibrium state.

Response:Thank you very much for improving our manuscript; In Figure 3d, we tried to peel off the bilayer hydrogels in swollen state, and in any case, the bilayer hydrogel could not be peeled off to obtain a complete NIPAM hydrogel layer. Thus, the bonding force between two layers was difficult to test. After destroying the NIPAM hydrogel layer, the interface of the bilayer hydrogel showed a rough state. These indicated that bilayer hydrogel with strong adhesive interface was successfully prepared in this work.

  1. Please mention what solvent was used.  ‘The double-layer hydrogel, AAm/AAc hydrogel and NIPAM hydrogel all in swelling equilibrium state were cut into 30 mm × 4 mm (length × width) strips…’

Response:Thank you very much for improving our manuscript; the solvent is water, which had been added in our revised manuscript.

  1. In Figure 5, which is AAc/AAm hydrogel and which is NIPAM? Please include details.

Response:Thank you very much for improving our manuscript; we had modified this part in our revised manuscript. Figure 5 (a) AAc/AAm and (b) NIPAM hydrogels in 0.6 mol/L NaCl, 0.6 mol/L CaCl2, 0.6 mol/L NiCl2, 0.6 mol/L AlCl3, 0.6 mol/L FeCl3, 0.6 mol/L CrCl3, and 0.6 mol/L CeCl3 solution after 12 h.

  1. In Figure 6, please draw a simple scheme of the device structure. What is the top layer, and what is the bottom layer?

Response:Thank you very much for improving our manuscript; we had modified this part in our revised manuscript. Figure 6 The simulative petal closure process of the four-leaf clover hydrogel actuator when immersing it into 0.6 mol/L AlCl3 (a) the NIPAM hydrogel layer is on top and the AAm/AAc hydrogel layer is on the bottom, or 0.6 mol/L FeCl3 (b) the AAm/AAc hydrogel layer is on top and the NIPAM hydrogel layer is on the bottom.

  1. The bilayer actuator was only demonstrated with one-time use, i.e. bending when soaked in metal ions. If wanted to be used as a controllable switch or smart device, from the bending to a non-bending state needs to be proved, it would add novelty to realize a reversible behavior.

Response:Thank you very much for improving our manuscript; The bilayer actuator showed a reversible behavior under thermal stimuli, while could not be recovered from the bending to a non-bending state under metal ion stimuli. However, the cost of the bilayer actuator was very low, which indicated that the bilayer actuator would be suitable used as a controllable switch or smart device for one-time use. We had modified the revised manuscript to emphasize this.

We have revised our manuscript as best as we could, according to your comments. We really hope our revised manuscript could be suitable for the publication in Polymers”. Thanks

Thank you very much for all your help and looking forward to hearing from you soon.

Best regards

Yours sincerely

Huilong Guo

Institute Of Biological And Medical Engineering, Guangdong Academy of Sciences Contact email: huilongguo@126.com

REFERENCES

  1. Beaudoin, G.; Lasri, A.; Zhao, C.; Liberelle, B.; De Crescenzo, G.; Zhu, X.-X., Making Hydrophilic Polymers Thermoresponsive: The Upper Critical Solution Temperature of Copolymers of Acrylamide and Acrylic Acid. Macromolecules 2021, 54 (17), 7963-7969.
  2. Fu, H.; Hong, X.; Wan, A.; Batteas, J. D.; Bergbreiter, D. E., Parallel Effects of Cations on PNIPAM Graft Wettability and PNIPAM Solubility. Acs Appl Mater Inter 2010, 2 (2), 452-458.
  3. Zhang, Y.; Furyk, S.; Bergbreiter, D. E.; Cremer, P. S., Specific Ion Effects on the Water Solubility of Macromolecules:  PNIPAM and the Hofmeister Series. Journal of the American Chemical Society 2005, 127 (41), 14505-14510.

Round 2

Reviewer 1 Report (Previous Reviewer 1)

The authors have mostly addressed my concerns. One remaining concern is about the data in Table 1. I would reduce the significant digits of the LA/LN to 3.

Author Response

Thank you very much for improving our manuscript;  we had reduced the significant digits of the LA/LN to 3 about the data in Table 1 of our revised manuscript. 

This manuscript is a resubmission of an earlier submission. The following is a list of the peer review reports and author responses from that submission.

Round 1

Reviewer 1 Report

In this work, the authors present the synthesis and characterization of bilayer hydrogels composed of acrylic acid, acrylamide and N-isopropyl acrylamide. While the system itself is not particularly novel, the authors present a rather systematic study on how the bilayer hydrogels behave in different solution environment, including temperature and ionic compositions. The authors conclude that the bilayer gels, responding to both the temperature and osmolality, can be a sensor material. 

In general the study is carried out with care, and the results are clearly presented. I would recommend publication of this manuscript upon the following points are addressed:

- In the synthesis, Rhodamine B was used to stain the second layer of the layered gel. How are the dyes harnessed to the gel network? There are some work in which the release of Rhodamine B from polyacrylamide hydrogels are used as representative of release kinetics of low molecular weight drugs (10.1134/S1063784218090141 for example). If any conjugation chemistry was used to harness the dye, that should be mentioned.

- A big factor determining the behavior of acrylic acid-containing hydrogels is the pH and the degree of deprotonation. How were the pH controlled during the polymerization and characterization steps? This is probably relevant to the salt series you performed, since these can be strong or weak acids.

- Have you characterized how deep the interpenetrating network forms? This is relevant to the device construction, since smaller devices are obviously more responsive to stimuli (due to faster kinetics) but smaller device construction is limited by the absolute depth of IPN formation.

Author Response

In this work, the authors present the synthesis and characterization of bilayer hydrogels composed of acrylic acid, acrylamide and N-isopropyl acrylamide. While the system itself is not particularly novel, the authors present a rather systematic study on how the bilayer hydrogels behave in different solution environment, including temperature and ionic compositions. The authors conclude that the bilayer gels, responding to both the temperature and osmolality, can be a sensor material. 

In general the study is carried out with care, and the results are clearly presented. I would recommend publication of this manuscript upon the following points are addressed:

We have revised our manuscript as best as we could, according to the reviewers' comments and the editor’ requirements. The detailed responses to comments are listed below. We really hope our revised manuscript could be suitable for the publication in Polymers”.

- In the synthesis, Rhodamine B was used to stain the second layer of the layered gel. How are the dyes harnessed to the gel network? There are some work in which the release of Rhodamine B from polyacrylamide hydrogels are used as representative of release kinetics of low molecular weight drugs (10.1134/S1063784218090141 for example). If any conjugation chemistry was used to harness the dye, that should be mentioned.

Response:Thank you very much for improving our manuscript; No special conjugation chemistry was used to harness the dye, but it could be seen in Figure 3a that after soaking in deionized water for 12h, the color of the hydrogel stained with Rhodamine B is still red, and the release of Rhodamine B from the hydrogel has no effect on the test of this experiment. Thus, Rhodamine B-stained hydrogels in our manuscript are feasible. We really hope the reviewer can agree with this.

- A big factor determining the behavior of acrylic acid-containing hydrogels is the pH and the degree of deprotonation. How were the pH controlled during the polymerization and characterization steps? This is probably relevant to the salt series you performed, since these can be strong or weak acids.

Response:Thank you very much for improving our manuscript; In order to be convenient and reliable for practical applications, the pH had not been controlled during the polymerization and characterization steps. We had emphasized this in our revised manuscript. And we believe this revised manuscript is suitable for the publication in “Polymers” We really hope the reviewer can agree with this. Thank you for your useful suggestions, more systematically investigation about the effects of different ion concentrations and different pH values on the shape deformation process will be reported in our future work.

- Have you characterized how deep the interpenetrating network forms? This is relevant to the device construction, since smaller devices are obviously more responsive to stimuli (due to faster kinetics) but smaller device construction is limited by the absolute depth of IPN formation.

Response:Thank you very much for improving our manuscript; The interpenetrating network on the interface had not been characterized in our revised manuscript. We checked some literature, and also consulted many experts; it is still difficult for us to characterize the interface interpenetrating network. However, interpenetrating network on the interface was mainly used to improve interfacial adhesion performance in this work. And the interface does not appear to be peeled off during the whole test process, metal ion and thermal responsive bilayer hydrogel actuator was successfully achieved in our work. Besides, crosslinking densities of the hydrogels, in-depth discussion for the results and discussion part had been added in our revised manuscript. We had revised our manuscript as best as we could, according to your useful suggestions. We really hope our revised manuscript could be suitable for the publication in “Polymers”.

Reviewer 2 Report

1. The manuscript should be rejected because the manuscript has inadequate and insufficient data to meet the criteria relevant for publication in a high-impact scientific Journal.

2. The manuscript lacks novelty and originality.

3.  Hydrogels based on acrylic acid (AAc), acrylamide (AAm), and NIPAM, along with their mutual combinations,  are convincingly the most extensively investigated hydrogels. Bilayer hydrogels and hydrogel responsiveness are the subjects of numerous investigations. NIPAM hydrogels are scholarly examples of therm-responsive materials.  Regarding that, the introduction part is inadequate and lacks relevant literature data.  Given statement  “And in Poly acrylic acid based hydrogels, coordination bond can be form between carboxylic acid and trivalent metal ion which will lead to a high degree of AAc polymer chain, resulting in shrinkage of the AAc based hydrogel networks “ is inadequately presented.

4. The obtained results are mainly given through the images and superficial descriptions without any scientific explanation.   

5. The author should carefully proofread their manuscript before the next submission because there are numerous errors and confusion throughout the manuscript. 

Author Response

  1. The manuscript should be rejected because the manuscript has inadequate and insufficient data to meet the criteria relevant for publication in a high-impact scientific Journal.

Response:Thank you very much for improving our manuscript; The structure characterization about crosslinking densities of the hydrogels and in-depth discussion for the results and discussion part had been added in our revised manuscript. We had revised our manuscript as best as we could, according to your useful suggestions. We really hope our revised manuscript with sufficient data and in-depth discussion could be suitable for the publication in “Polymers”.

  1. The manuscript lacks novelty and originality.

Response:Thank you very much for improving our manuscript; While the system itself is not particularly novel, we present a rather systematic study on how the bilayer hydrogels behave in different solution environment, including temperature and ionic compositions. To our best knowledge, this is the first work on the investigation of different metal ion responsive bilayer hydrogel actuator caused by shrinkage ratio of the two layers of the hydrogels immersed in different metal ion aqueous solutions. We really hope the reviewer can agree with the novelty and originality of our revised manuscript.

  1. Hydrogels based on acrylic acid (AAc), acrylamide (AAm), and NIPAM, along with their mutual combinations, are convincingly the most extensively investigated hydrogels. Bilayer hydrogels and hydrogel responsiveness are the subjects of numerous investigations. NIPAM hydrogels are scholarly examples of therm-responsive materials. Regarding that, the introduction part is inadequate and lacks relevant literature data. Given statement “And in Poly acrylic acid based hydrogels, coordination bond can be form between carboxylic acid and trivalent metal ion which will lead to a high degree of AAc polymer chain, resulting in shrinkage of the AAc based hydrogel networks “ is inadequately presented.

Response:Thank you very much for improving our manuscript; The second and third paragraphs of introduction part in our revised manuscript had been enriched according to your advice, and related literature to support the introduction had been added.

To address this issue, varieties of anisotropic structures, such as oriented structures 1, gradient structures2, patterned structures3-4, and bilayer structures5-12 have been investigated. Among them, bilayer structures are increasingly favored by researchers for their ease of preparation and design diversity. 13 Bilayer hydrogels will undergo asymmetric responsive shape-morphing behavior upon encountering external stimuli, thus leading to macroscopic bending, stretching, and twisting. And asymmetric osmotic flow of water and interfacial adhesion between the two layers, are crucial for hydrogel actuators.6, 10 In our previous work14, “C” or “S” shape actuator were prepared by applying a layer of thin hydrophobic adhesive film into the hydrophilic hydrogel film according to specific shape designs. The actuator could bend toward the hydrophobic film owing to asymmetric osmotic flow of water which was caused by the differences in hydrophilicity-hydrophobicity of the two layers. However, the interfacial adhesion was formed by physical adhesion which was not stable enough to undergo long time or repeated shape morphing process.

It is reported that a strong bilayer interface can be achieved through formation of semi-interpenetrating network structure at the interface of the two hydrogel layers.15 The semi-interpenetrating network can be formed by in situ polymerization of the second hydrogel layer on the prepared first hydrogel layer. During polymerization process of the second layer, some active reactants will penetrate into the first hydrogel network and then form a semi-interpenetrating network interface. NIPAM blocks may undergo phase transition and caused shrinkage of the hydrogel networks at a temperature above lower critical phase transition temperature (LCST). Thus, during the heating process, the bilayer hydrogel will bend toward the NIPAM layer owing to the asymmetric osmotic flow of water between the two layers. As known, specific interactions between ions, polymers, and internal water molecules will modulate the osmotic response depending on the kind of salts and the molecular structure of polymers. 16 And in Poly acrylic acid based hydrogels, coordination bond can be form between carboxylic acid and trivalent metal ion which will lead to a high degree of AAc polymer chain 17-22, resulting in shrinkage of the AAc based hydrogel networks. Therefore, it is expected that metal ion and thermal responsive hydrogel actuator can be achieved, owing to asymmetric osmotic flow of water between the two hydrogel layers under different stimuli, with a semi-interpenetrating network interface through in situ polymerization of the AAc based pregel layer on the prepared NIPAM hydrogel layer.

  1. The obtained results are mainly given through the images and superficial descriptions without any scientific explanation.

Response:Thank you very much for improving our manuscript; scientific explanation and in-depth discussion had been added in our revised manuscript.

  1. The author should carefully proofread their manuscript before the next submission because there are numerous errors and confusion throughout the manuscript.

Response:Thank you very much for improving our manuscript; We had improved the English with the help from professional editorial team (Owl Editing) and modified the manuscript carefully as best as we could.

References

  1. Kim, Y. S.; Liu, M.; Ishida, Y.; Ebina, Y.; Osada, M.; Sasaki, T.; Hikima, T.; Takata, M.; Aida, T., Thermoresponsive actuation enabled by permittivity switching in an electrostatically anisotropic hydrogel. Nat Mater 2015, 14 (10), 1002-1007.
  2. Tan, Y.; Wang, D.; Xu, H.; Yang, Y.; Wang, X.-L.; Tian, F.; Xu, P.; An, W.; Zhao, X.; Xu, S., Rapid Recovery Hydrogel Actuators in Air with Bionic Large-Ranged Gradient Structure. Acs Appl Mater Inter 2018, 10 (46), 40125-40131.
  3. Bowen, J. J.; Rose, M. A.; Konda, A.; Morin, S. A., Surface Molding of Microscale Hydrogels with Microactuation Functionality. Angewandte Chemie International Edition 2018, 57 (5), 1236-1240.
  4. Wang, Z. J.; Zhu, C. N.; Hong, W.; Wu, Z. L.; Zheng, Q., Cooperative deformations of periodically patterned hydrogels. Science Advances 2017, 3 (9), e1700348.
  5. Han, Z.; Wang, P.; Mao, G.; Yin, T.; Zhong, D.; Yiming, B.; Hu, X.; Jia, Z.; Nian, G.; Qu, S.; Yang, W., Dual pH-Responsive Hydrogel Actuator for Lipophilic Drug Delivery. Acs Appl Mater Inter 2020, 12 (10), 12010-12017.
  6. He, X.; Zhang, D.; Wu, J.; Wang, Y.; Chen, F.; Fan, P.; Zhong, M.; Xiao, S.; Yang, J., One-Pot and One-Step Fabrication of Salt-Responsive Bilayer Hydrogels with 2D and 3D Shape Transformations. Acs Appl Mater Inter 2019, 11 (28), 25417-25426.
  7. Hua, L.; Xie, M.; Jian, Y.; Wu, B.; Chen, C.; Zhao, C., Multiple-Responsive and Amphibious Hydrogel Actuator Based on Asymmetric UCST-Type Volume Phase Transition. Acs Appl Mater Inter 2019, 11 (46), 43641-43648.
  8. Li, J.; Ma, Q.; Xu, Y.; Yang, M.; Wu, Q.; Wang, F.; Sun, P., Highly Bidirectional Bendable Actuator Engineered by LCST-UCST Bilayer Hydrogel with Enhanced Interface. Acs Appl Mater Inter 2020, 12 (49), 55290–55298.
  9. Hou, K.; Nie, Y.; Mugaanire, I. T.; Guo, Y.; Zhu, M., A novel leaf inspired hydrogel film based on fiber reinforcement as rapid steam sensor. Chemical Engineering Journal 2020, 382, 122948.
  10. He, X.; Sun, Y.; Wu, J.; Wang, Y.; Chen, F.; Fan, P.; Zhong, M.; Xiao, S.; Zhang, D.; Yang, J.; Zheng, J., Dual-stimulus bilayer hydrogel actuators with rapid, reversible, bidirectional bending behaviors. Journal of Materials Chemistry C 2019, 7 (17), 4970-4980.
  11. Cheng, Y.; Ren, K.; Huang, C.; Wei, J., Self-healing graphene oxide-based nanocomposite hydrogels serve as near-infrared light-driven valves. Sensors and Actuators B-Chemical 2019, 298, 126908.
  12. Lu, H.; Wu, B.; Yang, X.; Zhang, J.; Jian, Y.; Yan, H.; Zhang, D.; Xue, Q.; Chen, T., Actuating Supramolecular Shape Memorized Hydrogel Toward Programmable Shape Deformation. Small 2020, 16 (48), 2005461.
  13. Xu, W.; Dong, P.; Lin, S.; Kuang, Z.; Zhang, Z.; Wang, S.; Ye, F.; Cheng, L.; Wu, H.; Liu, A., Bioinspired bilayer hydrogel-based actuator with rapidly bidirectional actuation, programmable deformation and devisable functionality. Sensors and Actuators B-Chemical 2022, 359.
  14. Guo, H.; Dai, W.; Miao, Y.; Wang, Y.; Ma, D.; Xue, W., Sustained Heparin Release Actuator Achieved from Thermal and Water Activated Shape Memory Hydrogels Containing Main-chain LC Units. Chemical Engineering Journal 2018, 339, 459-467.
  15. Xiao, S.; Zhang, M.; He, X.; Huang, L.; Zhang, Y.; Ren, B.; Zhong, M.; Chang, Y.; Yang, J.; Zheng, J., Dual Salt- and Thermoresponsive Programmable Bilayer Hydrogel Actuators with Pseudo-Interpenetrating Double-Network Structures. Acs Appl Mater Inter 2018, 10 (25), 21642-21653.
  16. Park, K. C.; Tsukahara, T., Expansion of Ion Effects on Water Induced by a High Hydrophilic Surface of a Polymer Network. Langmuir 2020, 36 (1), 159-168.
  17. Ma, C.; Wang, Y.; Cao, Z.; Zheng, J., Effect of Water Dehydration Treatment on the Structure and Mechanical Properties of Poly(AAc-co-AM)/Fe~(3+) Coordination Hydrogels. Polymer Materials Science & Engineering 2020, 36 (10), 109-113,119.
  18. Zheng, S. Y.; Yu, H. C.; Yang, C.; Hong, W.; Zhu, F.; Qian, J.; Wu, Z. L.; Zheng, Q., Fracture of tough and stiff metallosupramolecular hydrogels. Materials Today Physics 2020, 13, 100202.
  19. Tang, L.; Liao, S.; Qu, J., Metallohydrogel with Tunable Fluorescence, High Stretchability, Shape-Memory, and Self-Healing Properties. Acs Appl Mater Inter 2019, 11 (29), 26346-26354.
  20. Sthoer, A.; Adams, E. M.; Sengupta, S.; Corkery, R. W.; Allen, H. C.; Tyrode, E. C., La3+ and Y3+ interactions with the carboxylic acid moiety at the liquid/vapor interface: Identification of binding complexes, charge reversal, and detection limits. J Colloid Interf Sci 2022, 608, 2169-2180.
  21. Sung, W.; Krem, S.; Kim, D., Binding of trivalent ions on fatty acid Langmuir monolayer: Fe3+ versus La3+. J Chem Phys 2018, 149 (16), 163304.
  22. Reese, S.; Kaden, P.; Taylor, C. J.; Kloditz, R.; Schmidt, M., Structure and Thermodynamics of Eu(III) and Cm(III) Complexes with Glucuronic Acid. Inorganic Chemistry 2021, 60 (19), 14667-14678.

Round 2

Reviewer 2 Report

I didn't find any significant improvement in regard to my first review (recommended Reject), nor in general.